Chronic trace metals effects of mine tailings on estuarine assemblages revealed by environmental DNA

http://orcid.org/0000-0002-1838-4597 Bernardino Angelo F. 1 bernardino.ufes@gmail.com
Pais Fabiano S. 2
Oliveira Louisi S. 1
Gabriel Fabricio A. 1
http://orcid.org/0000-0002-4088-7457 Ferreira Tiago O. 3
Queiroz Hermano M. 3
Mazzuco Ana Carolina A. 1
1 Grupo de Ecologia Bentônica, Department of Oceanography, Universidade Federal do Espírito Santo , Vitoria, Espirito Santo , Brazil
2 Instituto René Rachou, FIOCRUZ/Minas , Belo Horizonte, Minas Gerais , Brazil
3 Escola Superior de Agricultura Luiz de Queiroz, Universidade de São Paulo , Piracicaba, São Paulo , Brazil
Moyer Craig
Electronic publication date: 2019 Nov 7
Publication date: 2019
Volume: 7
Electronic Location ID: e8042
Received 2019 Aug 27; Accepted 2019 Oct 16
Copyright: © 2019 Bernardino et al.
Copyright year: 2019
Copyright holder: Bernardino et al.
License: This is an open access article distributed under the terms of the Creative Commons Attribution License, which permits unrestricted use, distribution, reproduction and adaptation in any medium and for any purpose provided that it is properly attributed. For attribution, the original author(s), title, publication source (PeerJ) and either DOI or URL of the article must be cited.
License URL: https://creativecommons.org/licenses/by/4.0/

Keywords: Benthos, Meiofauna, Impacts, Pollution, Estuary, Samarco, Rio, Doce

Funding: Angelo F. Bernardino and Tiago O. Ferreira from Fundação de Amparo do Espirito Santo 77683544/2017, 81712405/2018 Coordenação de Aperfeiçoamento de Pessoal em Nível Superior CAPES and Conselho Nacional de Pesquisa e Desenvolvimento CNPq 301161/2017-8, 305996/2018-5 This work was funded by grants to Angelo F Bernardino and Tiago O Ferreira from Fundação de Amparo do Espirito Santo (77683544/2017, 81712405/2018), Coordenação de Aperfeiçoamento de Pessoal em Nível Superior CAPES and Conselho Nacional de Pesquisa e Desenvolvimento CNPq (301161/2017-8; 305996/2018-5). The funders had no role in study design, data collection and analysis, decision to publish, or preparation of the manuscript.

==============================
Mine tailing disasters have occurred worldwide and contemporary release of tailings of large proportions raise concerns of the chronic impacts that trace metals may have on the aquatic biodiversity. Environmental metabarcoding (eDNA) offers an as yet poorly explored opportunity for biological monitoring of impacted aquatic ecosystems from mine tailings and contaminated sediments. eDNA has been increasingly recognized to be an effective method to detect previously unrecognized small-sized Metazoan taxa, but their ecological responses to environmental pollution has not been assessed by metabarcoding. Here, we evaluated chronic effects of trace metal contamination from sediment eDNA of the Rio Doce estuary, 1.7 years after the Samarco mine tailing disaster, which released over 40 million m3 of iron tailings in the Rio Doce river basin. We identified 123 new sequence variants environmental taxonomic units (eOTUs) of benthic taxa and an assemblage composition dominated by Nematoda, Crustacea and Platyhelminthes; typical of other estuarine ecosystems. We detected environmental filtering on the meiofaunal assemblages and multivariate analysis revealed strong influence of Fe contamination, supporting chronic impacts from mine tailing deposition in the estuary. This was in contrast to environmental filtering of meiofaunal assemblages of non-polluted estuaries. Here, we suggest that the eDNA metabarcoding technique provides an opportunity to fill up biodiversity gaps in coastal marine ecology and may become a valid method for long term monitoring studies in mine tailing disasters and estuarine ecosystems with high trace metals content.

Introduction

Environmental assessment studies rely on accurate detection of biodiversity of an extremely diverse and small-sized benthic fauna. For decades, morphological methods are the basis to impact assessment (IA) protocols at the cost of neglecting an enormous number of meiofaunal species that could not be accurately identified (Bhadury et al., 2006; Fonseca et al., 2010). There have been considerable advances in recent years by applying DNA-sequence based techniques, commonly referred as metabarcoding, to identify and quantify meiofaunal biodiversity (Creer et al., 2010; Bik et al., 2012; Brannock et al., 2014). Metabarcoding techniques can now be applied to access levels of richness and spatial patterns of diversity on marine Metazoans with use of homologous genes (nuclear 18S rRNA), and help uncover the extreme high biodiversity of meiofaunal benthic assemblages (Fonseca et al., 2010). These modern approaches offer fast assessments of marine Metazoan meiofaunal assemblages and are particularly useful for the identification of new species in areas with poorly reported biodiversity, which may be of special interest in IA studies.

Estuarine coastal ecosystems offer an opportunistic case to evaluate biodiversity-environmental relationships through environmental DNA (eDNA) since many estuaries are widely impacted by pollutants with deleterious effects to benthic assemblages (Lotze et al., 2006; Chariton et al., 2015; Hadlich et al., 2018; Lana & Bernardino, 2018). The Samarco mine tailing disaster that occurred in Brazil on November 2015, released near 43 million m3 of tailings in the Rio Doce river basin, which were transported for over 600 km until reaching the estuary and the Atlantic Ocean (Carmo et al., 2017; Magris et al., 2019). The tailings severely impacted the Rio Doce riverine and estuarine ecosystems causing rapid sediment accumulation, burial and death of benthic organisms, and rapidly (1 and 2 days) increased sediment heavy metal accumulation by orders of magnitude from pre-impact conditions (Gomes et al., 2017). Although the released tailings had trace metal concentrations that were within the Brazilian legislation (Segura et al., 2016); the iron tailings deposited in the estuarine soils were heavily associated with trace metals which are potentially bioavailable given the redox conditions of estuarine soils (Queiroz et al., 2018).

Trace metal accumulation in coastal ecosystems are reported to be highly associated with changes in benthic assemblages and to increase human health risks due to potential bioaccumulation in food webs (Venturini, Muniz & Rodriguez, 2002; Muniz et al., 2004; Rainbow, 2007; Hauser-Davis et al., 2012). As a result, impact assessment studies that followed the Samarco disaster were also based on traditional morphological biodiversity assessments (Gomes et al., 2017). The potential chronic pollution effects in the Rio Doce estuary will likely demand long term monitoring programs for this ecosystem. To that end, technical and taxonomic expertise will be of key importance to monitor the estuarine biodiversity, but these efforts are typically limited to the macrofaunal and megafaunal benthic taxa. Therefore, monitoring this environmental disaster by increasing its biodiversity assessment to a broader range of cryptic and meiobenthic taxa may bring valuable information on the extension of impacts.

In this study, we used an eDNA metabarcoding approach to evaluate the benthic biodiversity in the Rio Doce estuary 1.7 years after the initial impacts of the Samarco disaster. We hypothesized that spatial patterns of chronic metal contamination in the estuary would be significantly associated with patterns of meiofaunal environmental taxonomic units (eOTUs), evidencing the potential use of this technique for long term impact assessment of the estuary. We targeted benthic meiofaunal eukaryote organisms by amplifying and sequencing the V9 hypervariable region of the 18S ribosomal gene from purified eDNA. In addition, sediment variables (particle size, organic carbon content) and trace metals concentrations were used to test for spatial changes in benthic assemblages in response to contamination levels in the estuary.

Materials and methods

Study site

The Rio Doce estuary (19°38′ to 19°45′ S, 39°45′ to 39°55′ W; Fig. 1), is located on the Eastern Marine Ecoregion of Brazil that has two well-defined seasons, dry winter (April–September) and wet summer (October–March), with an average monthly rainfall of 145 mm and temperatures of 24 to 26 °C (Bernardino et al., 2018; Bissoli & Bernardino, 2018). The Rio Doce estuary has been altered by historical human occupation, but ecosystem health of the estuary was poorly known before the Samarco disaster that occurred in November 2015 (Bernardino et al., 2016; Gomes et al., 2017). The initial impacts of the Samarco disaster in the estuary were reported by Gomes et al. (2017), and a standard monitoring of benthic assemblages and contamination levels were established in 2017 with a disaster-response program funded by Brazilian government agencies (Fapes, Capes and CNPq). The first monitoring campaign occurred in August 2017 (SISBIO sampling license N 24700-1), approximately 1.7 years after the initial impacts were observed in the estuary, when we aimed to quantify the potential chronic effects of trace metal pollution that were first observed in November 2015.

Figure 1 Map of the study site.

Map of sediment sampling stations at the Rio Doce estuary, Brazil in August 2017.

Sample collection and DNA isolation

Environmental DNA was obtained from two biological replicates of estuarine undisturbed surface (0–5 cm) sediments samples at 22 sites on the Rio Doce estuary in August 2017 (Fig. 1). The top 5 cm (~300 g wet weight) sediment was sampled with DNA-free sterile material and immediately frozen in liquid nitrogen. In the laboratory, all glassware was cleaned and autoclaved between samples to avoid cross contamination. Sediment samples were elutriated in DNA-free material to concentrate benthic metazoans and eDNA was extracted following protocols of Brannock & Halanych (2015), stored at −20 °C and sent to the Genomic Services Laboratory at Hudson Alpha Institute for Biotechnology (Huntsville, AL, USA) for metabarcoding sequencing. Briefly, the total DNA from 200 g (ww) of frozen sediments were extracted from each replicate separately with a Mobio PowerSoil (R) kit according to manufacturer’s protocol with a 2 min bead-beating step. DNA integrity was evaluated using electrophoresis on 1% agarose gels and DNA purity was assessed with a NanoDrop spectrophotometer (Thermo Fisher Scientific Inc., Waltham, MA, USA). Accurate DNA quantification was obtained using a Qubit® 3.0 Fluorometer (Life Technologies-Invitrogen, Carlsbad, CA, USA). Only 20 stations had enough bulk DNA after extraction and seven samples out of the expected 40 replicates did not yield high quantities of purified eDNA. In total 33 sediment eDNA samples from the Rio Doce estuary were then submitted to amplicon library preparation and Illumina sequencing (Table 1).

Table 1 Sediment, eDNA and environmental variables in the Rio Doce estuary.

Salinity, Sediment total organic matter (TOM, %), particle size (% sand), concentration of trace metals (Fe, As and Pb), and Number of sequence variant reads (reads SV) and richness per station. All data sampled in August 2017 or 1.7 years after the Samarco disaster. Trace metals averaged from N = 2 replicates (SE). N, number of eDNA replicate samples sequenced per station. Reads SV. Total marine/aquatic meiofaunal sequence variants.

Station (N)	Salinity	TOM (%)	%sand	Fe (mg.kg−1)	As (mg.kg−1)	Pb (mg.kg−1)	Reads SV	Total SV richness	
2 (2)	1.0	6.2	12	42,343 (2,468)	.3 (0.1)	56.9 (4.8)	256,072	32	
3 (2)	0.6	16.8	72	41,808 (1,278)	10.1 (14)	77.8 (2.7)	265,363	33	
4 (2)	0.3	2.1	90	33,681 (2,429)	4.5 (1.6)	173.3 (7.8)	359,718	40	
5 (2)	0.3	2.2	95	28,710 (3,686)	1.6 (2.2)	115.4 (2.8)	293,669	51	
7 (2)	0.4	2.1	64	36,142 (134)	0.1 (0.2)	74.5 (7.5)	101,127	23	
8 (2)	0.2	1.5	96	21,419 (3,212)	0.1 (0)	134.8 (5.8)	238,735	39	
9 (2)	1.0	1.9	91	28,155 (1,391)	28.8 (34.3)	111.4 (40.7)	254,335	44	
10 (2)	0.2	3.5	89	27,184 (227)	0.1 (0)	83.1 (5.9)	226,548	35	
11 (2)	0.2	5.2	70	43,116 (2,768)	0.1 (0)	67.3 (4.3)	272,299	34	
12 (1)	0.1	2.4	84	39,029 (11,713)	13.3 (16.2)	174.4 (28)	132,722	54	
13 (2)	0.2	13.8	91	54,983 (4,157)	3.9 (5.5)	117.3 (12.9)	236,707	53	
14 (1)	1.6	2.4	86	27,920 (7,793)	0.1 (0)	30.3 (11.8)	71,648	40	
15 (2)	3.7	6	85	34,532 (1,980)	16.7 (15.6)	78.2 (4.6)	320,192	41	
16 (1)	0.3	3.9	90	31,539 (1,001)	0.0 (0)	33.1 (2.1)	35,915	44	
17 (2)	0.1	1.7	90	21,191 (42)	2.1 (0.2)	192.9 (15.1)	54,355	50	
18 (2)	0.2	3.2	88	37,781 (1,120)	11.2 (2.7)	160.8 (6.2)	222,481	46	
19 (1)	0.4	2.3	93	36,244 (801)	3.7 (0.5)	118.0 (3.5)	103,987	31	
20 (1)	1.9	1.9	62	18,814 (94)	0.1 (0)	14.2 (1.5)	139,712	29	
22 (1)	1.3	10.2	91	24,501 (3,804)	0.1 (0)	16.0 (6.9)	69,713	16	
23 (1)	0.3	2.5	89	44,506 (1,079)	4.9 (2.2)	99.1 (7.6)	133,700	26	

Sediment samples were obtained for trace metals, grain size and total organic matter (TOM) analysis and frozen (−20 °C). Grain size was analyzed by sieving and pipetting techniques (Suguio, 1973). TOM content was quantified gravimetrically as the weight loss after combustion (500 °C for 3 h). In each station, metal contamination was evaluated from two independent replicate samples. For the total trace metal contents, ~1 g of dry sediment samples were digested by an acid mixture (HCl + HNO3 + HF; United States Environmental Protection Agency, 1996) in a microwave digestion system. Following digestion, concentrations of trace metals (Al, Ba, Cr, As, Fe, Zn, Mn, Pb, Cd, Co) in all samples were determined using an inductively coupled plasma optical emission spectroscopy (ICP-OES; Thermo Scientific—iCAP 6200).

Illumina sequencing and bioinformatic pipelines

eDNA samples were sent to the Genomic Services Laboratory at Hudson Alpha Institute for Biotechnology (Huntsville, AL, USA) for amplicon sequencing. The Eukaryotic-specific V9 hypervariable region of 18S SSU rRNA gene was amplified using primers Illumina_Euk_1391f forward primer [GTACACACCGCCCGTC] and Illumina_EukBr reverse primer [TGATCCTTCTGCAGGTTCACCTAC] (Caporaso et al., 2010). The V9 region has been previously shown to accurately identify eukaryotes from environmental samples and has an amplicon length suited to most commercially available Illumina platforms (Amaral-Zettler et al., 2009; Brannock & Halanych, 2015). Library size distribution was accessed using a 2,100 Bioanalyzer (Agilent, Santa Clara, CA, USA). Amplicons were sequenced on MiSeq (Illumina, San Diego, CA, USA) using the Reagent Kit v3 (300 bp PE).

Demultiplexed raw single-end reads for each sample were processed and analyzed using the 2018.8 distribution of the QIIME2 software suite to estimate the observed taxa across replicates (Bolyen et al., 2018). Fastq files were first imported as QIIME2 artifacts with the appropriate import plugin. Single-end reads were then denoised via dada2 (Callahan et al., 2016) with the dada2 denoise-single plugin, where the—p-trunc parameter was set to 270 to remove low-quality bases and the—p-trim was set to 20 to remove primer sequence. The taxonomic composition of the amplicon sequence variants, generated after running the dada2 plugin, were assigned using the machine learning Python library scikit-learn (Pedregosa et al., 2011). A pre-trained Naïve Bayes classifier, trained on Silva 132 database (Quast et al., 2013) clustered at 99% similarity, was downloaded from QIIME2 website (https://docs.qiime2.org/). The feature-classifier plugin was used to generate de classification results, and the taxonomic profiles of each sample were visualized using the taxa barplot plugin.

Statistical analysis

Only Metazoan variant calls were selected for ecological analysis. Comparisons of community composition were based on replicate averages of eOTU reads from benthic taxonomic groups. Benthic taxa were grouped for taxonomic comparisons into main taxa including Gastrotricha, Platyhelminthes, Nematoda, Annelida, Crustacea, Mollusca and Cnidaria. Other invertebrate taxa including Gnathostomulida, Micrognathozoa, Tardigrada, Rotifera and Bryozoa were grouped into ‘before Other invertebrates’. Unassigned or other taxa (e.g., Insecta) were represented as “before Other Metazoa”. Taxonomic (eOTUs) accumulation curves (Chao1) were compared across datasets by using: (i) full eOTU matrices (Table S1), (ii) dominant eOTUs with over 0.1% of total Metazoan reads (Table S2) and (iii) the baseline benthic morphological diversity from the Rio Doce estuary (Gomes et al., 2017). Chao one curves were based on presence-absence eOTU matrices integrated between replicates from each station and were estimated in Primer-e V6 (Clarke & Gorley, 2006).

The spatial consistency of metal contamination with benthic assemblage composition was tested with a Canonical Analysis of Principal coordinates (CAP; Anderson & Willis, 2003) complemented with multidimensional scaling (Anderson, 2001; McArdle & Anderson, 2001; Oksanen et al., 2013). Before the CAP analysis was run, the existence of highly correlated variables (trace metals) was assessed and trace metals with significant correlation with Fe contents were removed. The resulting multivariate analysis was only run with sediment contents of Fe, As and Pb, given their non-significant auto-correlations (Table S3). In addition, these trace metals (Fe and Pb) markedly increased (5 to 20-fold) in concentration with the impact (Gomes et al., 2017) and were often above the recommended limits within the Brazilian legislation (Guerra et al., 2017). Given that the concentration of other trace metals were highly correlated with Fe, Fe contents likely represent the overall effect of mine tailings deposited in the estuary (Queiroz et al., 2018).

The CAP was run based on presence or absence matrices with full Metazoan eOTUs and with the reduced assemblage composed of dominant reads (>0.1% of reads; Table S2). The CAP eOTU matrices were then compared with environmental (trace metal concentrations, sediment OM, % sand and salinity) spatial patterns based on Euclidean distances matrix to determine vectors that contributed to differences among samples (Mazzuco et al., 2019). Graphical and analytical processing were performed in R project (R Core Team, 2016) with the packages: ‘stats’ and ’vegan’ (Oksanen et al., 2013).

Results

The Rio Doce estuary exhibited low salinities at the time of sampling (0.1–3.7). Sediments were dominated by sand particles (>62% sand), with the exception of site two which showed less sand-sized particles (12%; Table 1). Sediment TOM varied from 1.5 to 16.8%, with the highest organic content at stations three, 13 and 22 (16.8, 13.8 and 10.2%; respectively). Several estuarine areas had TOM in a similar range of 2–6.2% (Table 1), and sediment organic content was significantly correlated to Fe content (Pearson r = 0.5043, p = 0.023; Table S4). The concentration of trace metals in the estuarine sediments also varied markedly along the studied area (Table 1). Fe concentrations ranged from 18,814 to 54,982 mg kg−1 and were highly correlated with several other trace metals including Al, Cd, Cr, Co, Cu, Mn and Zn (Tables S3 and S5).

We obtained a total of 9,836,039 sequence reads, of which 6,840,886 were of high quality. The number of sequence reads per station ranged from 35,915 (St 16) to 359,718 (St 4), with an average of 207,285 total sequence variants per station. Stations that had only one replicate sequenced had a lower (e.g., stations 14 and 16) or a similar number of reads (e.g., stations 19 and 20) of sites that had two replicates sequenced. On average, 55.4% of reads corresponded to aquatic or marine Metazoan taxa (Table S1). The eOTU richness per station ranged over three-fold from 16 to 54 eOTUs (Table 1). Assemblages were dominated by Nematoda (34 eOTUs), Platyhelminthes (19), Crustacea (18), Gastrotricha and Annelida (12 eOTUs each; Table S1; Fig. 2). Most sites had over 80% of sequence variant reads represented by two to three meiofaunal taxa, including the dominant Gastrotricha, Nematoda and Crustacea. The number of unassigned Metazoan taxa was large (>50%) at stations 16 and 17, whereas it remained less than 20% in most sites.

Figure 2 Benthic assemblage composition of the Rio Doce estuary.

Benthic meiofaunal assemblage composition based on eDNA samples from the Rio Doce estuary in August 2017.

The eDNA species accumulation curves did not reach an asymptote with addition samples suggesting an yet incomplete biodiversity characterization of the estuary (Fig. 3). Several eOTUs (N = 88) were represented by less than 0.1% of sequence variant reads. When we removed the eOTUs that had less than 0.1% of sequence reads, the species accumulation stabilized at 32 eOTUs with five to seven samples, with no additional gain of taxa. The species accumulation asymptote with dominant eOTUs was reached in about half the number of samples necessary in morphology-based studies (12–14 samples; Fig. 3). The number of meiofaunal eOTUs (eOTU richness) were largely uncorrelated to sediment grain size (p = 0.161), Fe (p = 0.647) and organic matter content (p = 0.6395; Table S4; Fig. 4).

Figure 3 Taxa accumulation curves from eDNA samples.

Taxa accumulation curves (Chao1 index) based on full eOTU matrices (eDNA, blue dotted line), dominant eOTUS (>0.1% sequence reads; eDNA_dom black dotted line) and on morphology-based macrofaunal pre-impact assessments (author’s data published on Gomes et al., 2017) in the Rio Doce estuary.

Figure 4 Correlation of eOTU richness with sediment Fe and TOM content.

Correlation of eOTU richness with sediment Fe and TOM content across all sampling stations in the Rio Doce estuary in August 2017.

The multivariate patterns of dominant meiofaunal (S = 32) eOTU composition were significantly related to Fe contents in sediments (F = 2.89, p = 0.018, Fig. 5; Table 2). The CAP axes 1 and 2 explained 44% and 21% of multivariate variability; respectively (Table 2). Fe contents in sediments was associated to the multivariate distribution of meiofaunal eOTUs including the Nematoda Mesodorylaimus nigritulus and Epitobrilus stefanskii, Harpacticoid copepods, the Platyhelminthes Cirrifera dumosa and Bothrioplana sinensis, and Ostracods (Chrissia dongqianhuensis). Monhysteridae and Desmodorida spp. nematode worms were negatively correlated to Fe concentrations (CAP1 score= −0.25 to −0.18). Pb and As contamination were not correlated to Fe concentrations in sediments and were not significantly associated with the meiofaunal multivariate composition (Table 2).

Figure 5 Multivariate analysis of assemblage composition and environmental filtering in the Rio Doce in August 2017.

Canonical analyses of principal coordinates (CAP) ordination of samples according to multivariate distribution of dominant eOTUS (>0.1% total SV reads) in the Rio Doce estuary. The strength and direction of environmental effects (Spearman correlation values with p < 0.5 in red) on biological assemblages is represented by arrows of variable size. Environmental variables were based on Table 1 (Fe, Pb, As, Salinity, TOM and %Sand). Taxa scores indicate OTUs mostly correlated to site differences. Proportion of variance explained by axis 1 and 2 are in parenthesis. Symbol numbers indicate sampling station.

Table 2 Results of the canonical analysis of principal coordinates.

Results of the Canonical Analysis of Principal coordinates (CAP) testing the contribution of sediment (TOM%, sand content), water salinity and concentrations of trace metals in sediments (As, Fe, Pb) to the multivariate distribution of meiofaunal (eDNA) assemblages in the samples from Rio Doce estuary. Spearman correlation values for each sediment variable are described for in CAP axis one and two. proportion of variability explained by CAP axes are highlighted, F for statistic, significant results (p < 0.05) are in bold.

	All eOTUS (N = 123)	Dominant eOTUS (N = 32)	
	axis 1
0.33	axis 2
0.29	F	p	axis 1
0.44	axis 2
0.21	F	p	
Salinity	−0.63	0.09	1.24	0.223	−0.30	−0.71	1.60	1.77	
OM	−0.55	−0.18	0.89	0.485	−0.33	−0.36	0.71	0.592	
Sand	0.71	0.26	1.01	0.413	0.37	0.40	1.17	0.272	
As	0.02	0.26	0.63	0.815	0.39	0.31	0.79	0.506	
Fe	−0.29	0.46	1.45	0.135	0.43	−0.51	2.89	0.018	
Pb	0.70	−0.08	1.11	0.303	0.19	0.28	1.59	0.160	

Discussion

Our study demonstrates that environmental DNA can be an effective method to indicate chronic contamination effects on benthic assemblages of the Rio Doce estuary, supporting our hypothesis. This first eDNA survey in the Rio Doce estuary also revealed a previously unrecognized benthic biodiversity, even with significant impacts by trace metal levels 1.7 years after the initial impacts. Although there is no baseline eDNA assessment from the Rio Doce estuary, the impacted sediments potentially supported over 32 dominant meiofaunal taxa (eOTUs), with a spatial distribution significantly related to Fe (and correlated metals) contamination.

The Rio Doce eDNA composition was similar to other estuarine and marine sediments assessed by metabarcoding methods (Fonseca et al., 2010; Faria et al., 2018). Nematoda, Gastrotricha and Crustaceans were highly dominant in the estuary with local changes in relative abundance across sites sampled. The marked spatial variability in assemblage composition within the estuary indicates that benthic assemblages were spatially structured; which is a similar pattern commonly observed in morphology-based assessments. Environmental filtering in benthic assemblages may result from a combination of sediment and water variables, with grain size, salinity and food availability being critical to species turnover and replacement in estuarine benthos (Menegotto, Dambros & Netto, 2019). Although our study design does not allow for an hierarchical spatial analysis of variables that determined the observed environmental filtering, the detection of spatial variance in assemblages from eDNA samples suggests that the biodiversity assessment is likely representing living benthic organisms instead of predominantly ancient or allochthone DNA. There is now strong evidence supporting that eDNA techniques can detect complex spatial variability in estuarine and coastal marine ecosystems (Chariton et al., 2015; Faria et al., 2018); and our data additionally supports its use to biodiversity assessment in a heavily impacted estuary.

Most eOTUs represented new occurrences for the estuary, but yet with several unassigned taxa, stressing the complementarity value of molecular and morphological approaches to ecological and impact assessment studies (Leasi et al., 2018). We recovered a total of 123 environmental OTUs (eOTUs) in the Rio Doce estuarine sediments, increasing by over 20-fold the previous richness of benthic taxa based on morphological identifications (Gomes et al., 2017). The species accumulation curves did not reach an asymptote with addition of eDNA samples, and most eOTUs (N = 88) were represented by less than 0.1% of sequence variant reads, suggesting an yet incomplete biodiversity assessment of the Rio Doce estuary even with high levels of trace metals. However, estuaries are highly connected to continental and marine ecosystems and it is unlikely that species accumulation curves would reach an asymptote with a single biodiversity assessment (Chariton et al., 2015; Nascimento et al., 2018). The rapid increase and stabilization of the number of dominant meiofaunal OTUs with the addition of new samples suggests a reasonable beta-diversity assessment of the Rio Doce estuary with the effort taken. Sites that had only one sequenced replicate due to low DNA stocks attained similar or lower OTUs richness if compared to other stations, but the sediment volumes used in this study (>200 g) were well over the necessary to avoid technical bias in the detection of Metazoan diversity (Brannock & Halanych, 2015; Nascimento et al., 2018).

This single eDNA survey was efficient in assembling benthic meiofaunal assemblages in the Rio Doce estuary. The species accumulation curves indicate that half of the sampling effort would be necessary to characterize the dominant meiofaunal groups in the estuary if compared to the species accumulation rate of previous morphological assessments. eDNA metabarcoding can be more efficient at characterizing marine taxa (Lobo et al., 2017), and our data supports its use on long term studies where taxonomic and technical limitations cannot be controlled (Bista et al., 2017). The lack of controlled methods may be a crucial problem to the biodiversity monitoring that followed the Samarco disaster on the Rio Doce, given the extremely large scale and diversity of impacted ecosystems. It is estimated that over 2,000 ha of terrestrial, limnetic and estuarine ecosystems along the Rio Doce basin were directly impacted by the disaster (Carmo et al., 2017); with additional potential effects on nearby coastal zones (Magris et al., 2019). Given the large area, the diversity and natural complexity of ecosystems to be monitored, it is likely that the number of biological samples needed to reach reasonable statistical power to detect biodiversity impacts would be prohibitive (Fairweather, 1991). Therefore, the massive amount of data obtained in metabarcoding techniques could have a profound contribution to environmental monitoring in this scenario, which would also increase dramatically the discovery of cryptic species on a range of aquatic and terrestrial ecosystems.

Multivariate analysis revealed that Fe contents (and other correlated metals) are partially structuring spatial patterns of dominant benthic meiofaunal assemblages in the Rio Doce estuary 1.7 years after the disaster. The sediment Fe contents were significant predictors of changes in dominant meiofaunal eOTUs including nematodes, copepods, ostracods and flatworms. These groups corresponded to over 2/3 of meiofaunal OTUs in the estuary and revealed that trace metal contents are driving spatial patterns of the Rio Doce estuarine biodiversity. Our data suggest that benthic assemblages were highly sensitive to chronic metal contamination in polluted estuaries, and partially explains a lower effect of sediment grain size and organic matter on local meiofauna (e.g., Faria et al., 2018; Menegotto, Dambros & Netto, 2019). This could indicate that the Rio Doce estuarine assemblages were strongly impacted after the disaster through the exclusion of intolerant species, although we lack baseline eDNA to fully support that hypothesis.

The extremely high Fe contents allied to covariance of several potentially toxic trace metals that are adhered to iron oxides present in the tailings strongly suggest that the tailings have led to major changes in the estuarine benthic biodiversity since the initial impact (Queiroz et al., 2018). The Rio Doce basin was previously polluted by historical mining and urban activities, so the trace metals that rapidly accumulated in the estuary were likely transported downstream attached to Fe oxides from the released tailings. The initial impacts in the Rio Doce estuary were observed immediately with the arrival of tailings, which led to significant (2–20 times) increases in sediment Fe, Mn, Cr, Ni, Cu, Zn and As (Gomes et al., 2017). Sediment concentrations of Fe, Pb and selected trace metals in the Rio Doce in August 2017 continued to be 2–20 times higher compared to preserved (Piraquê-Açu-Mirim estuary) or polluted estuaries such as the Vitoria Bay, located in a major metropolitan and industrial area ~100 km to the south (Hadlich et al., 2018).

The statistical lack of As and Pb effects on the multivariate distribution and composition of meiofaunal assemblages have important implications for future environmental monitoring in the estuary. One plausible cause is that not all elements that are accumulated in the sediments are bioavailable and have toxicity to the estuarine biota. Given the amplitude of trace metals accumulated in the Rio Doce sediments since the mining tailing impacts that occurred in 2015, it is very likely that a combination of these contaminants lead to further changes in the estuarine benthos. The sediment concentrations of Pb in August 2017 were over 20 times higher than baseline values (Gomes et al., 2017); and several other trace metals also increased with time since the impact. Queiroz et al. (2018) reported a significant correlation between Fe, Pb and other trace metals in tailing deposits after the initial impacts in the Rio Doce estuary in 2015. The iron oxides from tailings deposited in the estuary have a strong capacity of metal retention (Cornell & Schwertmann, 2003; Yin et al., 2016); and they are likely to be released due dissimilatory iron reduction under estuarine conditions (Bonneville, Behrends & Van Cappellen, 2009; Queiroz et al., 2018; Xia et al., 2019). Although anoxic estuarine soils favor the formation of sulfides (e.g., Pyrite; AVS) which have strong affinity and role in the chelation of metals (Machado et al., 2010; Nóbrega et al., 2013), the Rio Doce estuary contrast to several other estuaries due to low salinity, low tidal influence and an apparent limited sulfate availability and sulfate reduction that reduce the formation of sulfides (Queiroz et al., 2018). As a result, the observed relationship of meiofaunal assemblages with Fe contents (and other trace metals) suggest that the tailings have some toxicity to benthic organisms even though a number of contaminants may not achieve alarming concentrations.

The effects of trace metal contents on the Rio Doce benthic assemblages resemble impacts in other areas that are highly polluted with trace metals, but these effects could be confounded with the constant environmental changes that typically occur in these ecosystems (Krull et al., 2014; Martins et al., 2015). Our approach of selecting dominant meiofaunal OTUs to multivariate analysis led to positive detection of Fe contents effects and associated trace metals. This approach was justifiable given that we detected 88 eOTUs with less than 0.1% of sequence variant reads, which could be potentially associated with allochthone DNA from connected river or ocean ecosystems and would not be under influence of local contaminants. The use of indicator taxa or functional groups to eDNA biodiversity assessment studies is becoming practice in ecological studies (e.g., Bista et al., 2017) and our approach offers an important methodological approach for detection of trace metals effects in aquatic biota that need to be further investigated in other case studies.

Conclusions

In conclusion, our eDNA survey of benthic meiofaunal assemblages in the Rio Doce estuary detected environmental filtering with strong influence of contamination by Fe and other correlated trace metals, supporting chronic mine tailing impacts in the estuary. Our study is also in agreement with previous assertions that ecological inferences from eDNA analysis may increase the performance of biodiversity assessments in marine ecosystems by capturing a range of cryptic taxa, thus greatly improving current short and long-term impact assessment studies. The use of eDNA to the Samarco mine tailing disaster would benefit monitoring assessments with standard techniques and dramatically increase our knowledge of the biodiversity of cryptic aquatic species. The continued sampling and monitoring would also increase the precision of the eDNA assessments, especially if allied to detailed morphological work.

Supplemental Information

Supplemental Information 1 List of marine eOTUs of aquatic Metazoan taxa recovered from sediment eDNA samples from the Rio Doce estuary.

Note: unassigned species identities are indicated as notID.

Click here for additional data file.

Supplemental Information 2 Dominant meiofaunal OTUs with >0.1% of sequence variant reads included in multivariate analysis.

Click here for additional data file.

Supplemental Information 3 Results of paired spearman correlation analysis of trace metals concentrations in sediment samples from the Rio Doce estuary in August 2017.

Variables were considered co-variating when the correlation coefficient (Pearson’s r) was not equal to 0 with p ≤ 0.05. Significant results (p < 0.05) are in bold.

Click here for additional data file.

Supplemental Information 4 Canonical correlation values (Pearson r and p values) among eOTUs and sedimentary Fe, Sand and TOM and among co-variates (Fe, Sand and TOM).

*Indicates significant correlations at the 0.05 level (two-tailed).

Click here for additional data file.

Supplemental Information 5 Trace metal concentrations (mg kg−1) in sediments at each sampling site of the Rio Doce estuary in August 2017.

Click here for additional data file.

Supplemental Information 6 Number of sequence variant reads from Metazoan taxa from the Rio Doce estuary in August 2017.

Distributed across sites sampled.

Click here for additional data file.

We thank students that helped with field sampling.

Additional Information and Declarations

Competing Interests

Author Contributions

Field Study Permissions

Data Availability

The authors declare that they have no competing interests.

Angelo F. Bernardino conceived and designed the experiments, performed the experiments, analyzed the data, contributed reagents/materials/analysis tools, prepared figures and/or tables, authored or reviewed drafts of the paper, approved the final draft.

Fabiano S. Pais performed the experiments, analyzed the data, contributed reagents/materials/analysis tools, prepared figures and/or tables, authored or reviewed drafts of the paper, approved the final draft.

Louisi S. Oliveira performed the experiments, analyzed the data, prepared figures and/or tables, authored or reviewed drafts of the paper, approved the final draft.

Fabricio A. Gabriel performed the experiments, analyzed the data, prepared figures and/or tables, authored or reviewed drafts of the paper, approved the final draft.

Tiago O. Ferreira performed the experiments, analyzed the data, contributed reagents/materials/analysis tools, authored or reviewed drafts of the paper, approved the final draft.

Hermano M. Queiroz performed the experiments, analyzed the data, authored or reviewed drafts of the paper, approved the final draft.

Ana Carolina A. Mazzuco performed the experiments, analyzed the data, prepared figures and/or tables, authored or reviewed drafts of the paper, approved the final draft.

The following information was supplied relating to field study approvals (i.e., approving body and any reference numbers):

Field sampling was approved by the SISBIO-IBAMA (sampling license N 24700-1).

The following information was supplied regarding data availability:

The raw data are available in the Supplemental Files.

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
