# Peer review of "Chronic trace metals effects of mine tailings on estuarine assemblages revealed by environmental DNA"

_PeerJ, doi:10.7717/peerj.8042_

## Round 0.1 · original submission · Minor Revisions

Please carefully consider each of the reviewers comments and then resubmit your revised manuscript.

Reviewer 1 ·

Basic reporting

The manuscript is well writen and clear, and proposes the use of eDNA as indicator of meiofaunal community, in order to observe effects caused by metals related to mining tailings in an estuarine region. In general, the objective is well defined, the results are properly presented and the discussion is well organized and address the aspects that attend the objectives. Some details can be provided on the approach used, and some parts of the discussion can be improved, based on the results or on a better exploration of the literature.

Experimental design

The experimental design is mostly ok. Regarding the geochemistry, the use of HF (together with other acids) to digest the sediment samples can cause an overestimation of the contamination levels, because it includes elements from the crystallin matrix.

Validity of the findings

Most of the results are ok. Some details could be provided, as for example the influence on the grain size or TOM on the benthic diversity and metals. Some caution should be taken when interpreting the chemistry data (as commented above).
Besides, to my best knowledge, the Rio Doce river basin presents a history of contamination by metals and pesticides, thus the benthic community probably is influenced by such hystorical factor, together with the contamination associated with the mining tailings. This should be discussed in more detail and include the uncertainty on determining if the current benthic condition is due to the disaster or to a combination of different contamination processes.
I also would like to see a more detailed comparison between classic morphological based benthic diversity and that provided by eDNA. How congruent are they?

Additional comments

Specific comments
Line 47 and on: You could provide more details on these techniques
Lines 129-131: Acid digestion with HF causes the extraction of elements related to the crystallin matrix and consequently may produce overestimation of the concentrations of bioavailable metals.
This fact may have implications for the results' interpretation.
Line 138: which are the advantages and limitations of using the 18S SSU rRNA gene?
Lines 196-200: How did metals concentrarions correlate with sediment textural classes and TOM?
Since sediment grain size distribution and TOM are important factor to condition the distribution of benthic species and metals, knowing such correlations would be hepful.
Line 253 and on: One would like to see the relationship between eOTUs and some regular biotic variables, such as species richness and/or diversity. Have you tried to do such comparisons?
Line 291: Consider the river basin (and I guess, the estuary as well) was previously contaminated by metals (mainly from mining activities located along the river basin)
Lines 299-304: As I mentioned, the sediments from the river basin were previouly contaminated by metals, thus some hystorical adaptation/resistance could be expected in native organisms.
Lines 307-308: this is quite possible, especially regarding the As, since the acid digestion used HF.
Lines 311-312: Is Pb (and other elements mentioned) related with tailings composition?
Lines 312-315: However, it is widely known that acid volatile sulfides become predominant in hypoxic (or anoxic) sediments and they are efficient chelators of metals. I suggest the authors to search some literature on the AVS-SEM ratios and their relationship with bioavailability and toxicity.
Line 316 and on: As mentioned, Fe occurs along the Rio Doce basin, prior to the disaster, possibly as a chronic consequence of the mining activities upstream. Additional amounts of Fe were brought to the system due to the mining tailings. Therefore, it is not possible to affirm that metals were due to the tailings only; instead, a certain fact is that metal contamination is mainly due to mining exploration (chronic contamination by the mining activities plus the residues from the disaster)
Line 322 and on: I agree with you. However, since you have information on the taxonomic distribution based on both eDNA and classic morphological characteristics, you could use the comparison to justify or corroborate the choice of using eDNA to estimate biodiversity in the studied case.
Line 345: no funding agency? no institutional support? no other support?

·

Basic reporting

no comment

Experimental design

no comment

Validity of the findings

no comment

Additional comments

This study investigated the effects of trace metals associated with mine tailings disaster in Rio Doce estuary area on the aquatic biodiversity using environmental metabarcoding (eDNA). Overall the manuscript is of great quality, well written, well structured and of interest. The main conclusion from the presented results is “We detected environmental filtering on the meiofaunal assemblages and multivariate analysis revealed strong influence of Fe contamination, supporting chronic impacts from mine tailing deposition in the estuary.” However, Fe is likely not the only one responsible for the observed environmental filtering, and all the other trace metals analysed and highly correlated with Fe should be considered in this result/phrasing, as some of those metals are much more toxic than Fe. As such, while it is understood that only Fe was used in the multivariate analysis, as the representative of the “same signal” and correlated with Al, Cd, Cr, Co, Cu, Mn and Zn, I advise the co-authors to rephrase the sentences when mentioning the effects of Fe by something like effects of Fe-group or Fe+other metals (?).
In addition, and as mentioned by the authors that there is a lack of environmental data for the Doce River, it is advised to present the concentrations of Al, Cd, Cr, Co, Cu, Mn and Zn per sampling site, adding on to table S1 for example.
The discussion could also benefit from a brief discussion on the concentration of metals found in this study and how they compare to non-impacted estuaries nearby, or to highly polluted areas, if such data exist… if not at least mention that fact, to give an idea to the reader whether the presented concentrations are much higher or not.

Throughout the manuscript, including table legends, the italics formatting is missing from p –values, r, or species names. Also superscript and lowerscript formatting is missing (e.g. lines 126, 130, 221, 224-226).
Line 178: Verify if reference Gabriel et al., in review is already published or remove it.
Line 196-198: This sentence is not necessary - The concentration of trace metals in the estuarine sediments also varied markedly along the studied area. - as the subsequent sentence says the same.
Line 332: The first sentence of the conclusion should directly state what in your results support your hypothesis that the impacts of tailings affect the benthic estuarine assemblages. Or a sentence similar to that in the abstract in lines 32-34 (but adding the other trace metals to Fe!).

I recommend the publication of this article in PeerJ, after addressing the minor revisions mentioned above.

---

## Round 0.2 · accepted · Accept

Thank you for such detailed attention to the reviewer's comments, which I too agree have made this a better manuscript.